# LEARNING RANDOMIZED ALGORITHMS
# WITH TRANSFORMERS

**Johannes von Oswald**[*1], **Seijin Kobayashi**[*1,2], **Yassir Akram**[*2], **Angelika Steger**[2]
[1] Paradigms of Intelligence Team, Google
[2] Department of Computer Science, ETH Zürich
[*] Equal contribution, order determined randomly
{jvoswald, seijink}@google.com, {yakram, asteger}@ethz.ch

## ABSTRACT

Randomization is a powerful tool that endows algorithms with remarkable properties. For instance, randomized algorithms excel in adversarial settings, often surpassing the worst-case performance of deterministic algorithms with large margins. Furthermore, their success probability can be amplified by simple strategies such as repetition and majority voting. In this paper, we enhance deep neural networks, in particular transformer models, with randomization. We demonstrate for the first time that randomized algorithms can be instilled in transformers through learning, in a purely data- and objective-driven manner. First, we analyze known adversarial objectives for which randomized algorithms offer a distinct advantage over deterministic ones. We then show that common optimization techniques, such as gradient descent or evolutionary strategies, can effectively learn transformer parameters that make use of the randomness provided to the model. To illustrate the broad applicability of randomization in empowering neural networks, we study three conceptual tasks: associative recall, graph coloring, and agents that explore grid worlds. In addition to demonstrating increased robustness against oblivious adversaries through learned randomization, our experiments reveal remarkable performance improvements due to the inherently random nature of the neural networks' computation and predictions.

## 1 INTRODUCTION

Randomization is inherent to nature and an important ingredient in numerous scientific fields. In computer science, for example, randomness can be a powerful theoretical and practical tool to design algorithms (Papadimitriou, 1994). However, understanding how, if, and when randomization can be beneficial for algorithms is neither obvious nor intuitive. Nevertheless, randomness is by now a well-established and widely used concept to design powerful and often strikingly simple algorithms (Motwani & Raghavan, 1995; Rabin, 1980). In particular, randomization is known to be crucial to obtain algorithms performing well in game-theoretical adversarial settings, cf. rock-paper-scissors as a simple example. Here, randomization is essential as otherwise players will get exploited from a clever opponent. Within algorithmic design, furthermore, randomized algorithms are often surprisingly simple to implement. While base versions often have a non-satisfactory high failure probability, simple repetition and majority voting strategies typically allow to enhance overall success probability drastically at comparably low cost, see our illustrative example below and Appendix E for a short background. In this paper we combine the power of randomization and deep learning, in particular transformer models Vaswani et al. (2017), and show that powerful randomized algorithms within transformers are discovered when optimized on adversarial objectives. These transformer algorithms are significantly more robust against oblivious adversaries and dramatically outperform deterministic strategies through majority voting.

To set the stage, let's consider the example of associative recall to build up our intuition, illustrated in Figure 1. Assume a simple computer system with memory size $M \cdot d$ bits needs to save $N$ vectors, each of $d$ bits. These vectors are denoted as *values* $v_i = [v_{i1}, \ldots, v_{id}] \in \{0, 1\}^d$ which are associated with a unique one-hot binary *key* $k_i = [k_{i1}, \ldots, k_{iN}]$ with $k_{ij} = \mathbb{1}_{i=j}$. The aim of the computer system is to retrieve the correct value vector if queried with some key $k_i \in \{k_1, \ldots, k_N\}$ after

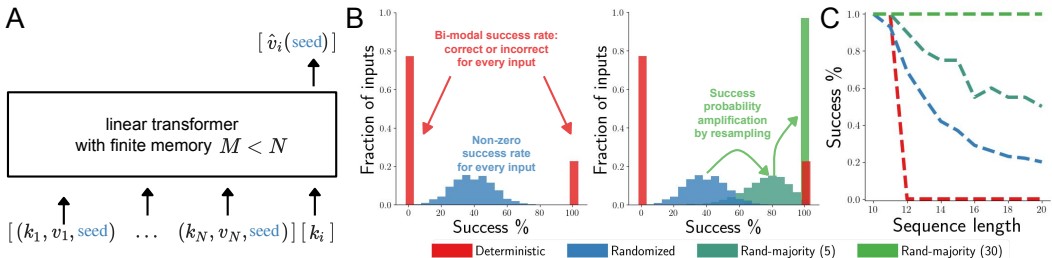

Figure 1: **Solving associative recall tasks - the randomized way**. **A)**: A causal transformer model, with linear self-attention layers, is trained to remember $N$ value vectors each associated with a unique key. Given finite memory of size $M < N$, the model needs to decide which data to memorize in order to return the correct value when queried with some $k_i$. An algorithm that deterministically chooses what to remember will perform poorly as simple adversarial strategies will break retrieval. On the other hand, randomly deciding what to store protects the algorithm against such worst-case scenarios. To have a transformer learn such a randomized strategy we train it with an adversarial objective and , furthermore, diverge from usual machine learning setups and provide randomness, denoted as a seed, as an additional input to the model. **B)**: Histogram showing the fraction of possible inputs, all of length $N = 20$, with varying recall success rates. *Left*: We train two different models: 1) a deterministic one where we fix the input seed and 2) a randomized one with varying input seed. While the deterministic model consistently fails on some fraction of the inputs, the randomized model stores in memory varying parts of the sequence leveraging the provided seed, thus succeeding with non-zero probability on all possible recall tasks. *Right*: At inference time, majority voting can be used on the randomized model evaluated on several seeds, thus amplifying the success rate. Enough seeds leads to optimal recall. **C)**: Recall success rate for worst-case sequences of each model when trained on various input lengths. For larger sequence lengths, the deterministic model fails to recall on adversarial inputs and the worst case success rate thus drops to zero. On the other hand, randomized models have a non-zero success rate on all inputs. Strikingly, when amplifying the success probability with majority voting, the randomized model succeeds in recalling virtually all inputs.

observing the whole sequence of $N$ value vectors. Crucially, if the memory of the system is not large enough i.e. $M < N$, the system needs to decide which data to store in memory and which data to disregard, assuming no compression is feasible. This problem is a variant of the well-known paging problem in computer science (Sleator & Tarjan, 1985a) for which randomized solutions exist, see Appendix E.3 for a short description. The problem is also a well-studied problem for transformers (Vaswani et al., 2017) and transformer architecture variants with the goal to assess retrieval capabilities (Schlag et al., 2021; Arora et al., 2023; Jelassi et al., 2024).

Let us first assume that the computer system or transformer implements a deterministic strategy. For example, it always saves the first $M$ elements in the sequence until memory capacity is reached. Given this knowledge, there is a simple adversarial strategy that breaks retrieval: query the system with keys $k_j$ where $j > M$. Note that a similar retrieval failure can be observed in large-scale language models where this test is known as finding a *needle in a haystack* (Liu et al., 2023b). Consider now that the transformer model is given an additional input, namely a random seed, and that based on the seed, it chooses uniformly at random which data to store. Then, every query will break retrieval (only) with probability $1 - \frac{M}{N}$. Therefore, in stark contrast to a deterministic system, there exists no input sequence on which the system will do specifically bad. In particular, no adversarial strategy can make the system fail consistently. In other words: randomization improves worst-case behavior, see Figure 1 B & C. Strikingly, randomization can enhance success rates of our trained transformers by repetition. Simply taking the majority vote among $m$ predictions computed on different seeds boosts performance to a perfect success rate far beyond the deterministic transformer counterpart - despite the same network capacity and training objective.

## 2 THEORETICAL CONSIDERATIONS

We start by presenting some well-known theoretical results providing a more thorough description of when and why randomized algorithms can be beneficial compared to deterministic ones. This will lead us to define a simple training objective, closely related to robust optimization, that enforces the

implementation of powerful randomized algorithms within deep neural networks like transformers[*]. We then verify experimentally, in the following section, that this is indeed the case and show that transformer models do implement deep randomized algorithms in practice after training.

To this extent, we introduce some notation and definitions. We denote our model, in this paper a parametrized transformer, by $A_\theta(x, r)$ with input $x \in \mathcal{X}$ and some randomness (or seed) $r$ from a set $\mathcal{R}$. We will fix distributions over the input space $\mathcal{X}$ and the seeds $\mathcal{R}$ of the transformer model. Specifically, we define two random variables $X$ (over $\mathcal{X}$) and $R$ (over $\mathcal{R}$). For any parameter $\theta$, $x \mapsto A_\theta(x, R)$ is a randomized transformer model , which we also call randomized transformer algorithm. We provide the transformer models in the following with a *random seed encoding* (RSE), similar to positional encodings common in transformer models. This encoding provides the randomness to the transformer and is usually a vector of noise, such as random bits, which we concatenate to the input tokens. Note that optimization can therefore ignore $R$, by setting, for example, the appropriate weights in the first layer to zero resulting in a deterministic transformer. When there exists $x \in \mathcal{X}$ such that the function $r \mapsto A_\theta(x, r)$ is not constant, we say that our transformer is *properly random*. For any fixed input $x \in \mathcal{X}$ and $r \in \mathcal{R}$, the loss is determined by $L(x, A_\theta(x, r)) \geqslant 0$. For notational simplicity, we will drop the dependence of the loss on $x$ in the following whenever it is clear from the context. The performance of the randomized transformer on the input $x$ is its expected loss, i.e. $\mathbb{E}[L(A_\theta(x, R))]$. We now review well-known results from the randomized algorithms literature and analyze when randomization is advantageous compared to determinism and vice versa. This analysis will lead to an optimization procedure from which we expect that the random source in the transformer's input is leveraged to reach lower loss values.

## 2.1 EXCESSIVE MODEL CAPACITY WILL NOT ENFORCE RANDOMNESS

A first necessary condition for randomness being beneficial is that the model capacity doesn't allow to achieve optimal performance without using randomness i.e. ignoring it in the input. More precisely, if there exists $\theta, r \in \mathcal{R}$ that perfectly fits the data, i.e. such that $A_\theta(x, r) \in \arg\min_y L(x, y)$ for all $x \in X$, then randomness has no incentive to be leveraged. Whether the optimization algorithm will discover this $\theta$ is however not guaranteed. We now continue by showing that commonly used expected risk minimization or empirical risk, in practice, should not lead to randomization.

## 2.2 RANDOMIZATION IS NOT BENEFICIAL IN EXPECTATION

To see that randomization is not beneficial in expectation, and therefore unlikely to be leveraged by optimization to reach lower loss, we discuss Yao's Minimax Principle (Yao, 1977). This is a pivotal concept across various domains like game theory and randomized algorithms Neumann (1928). In the latter, it provides a framework to understand how randomized algorithms fare in adversarial environments. It can be articulated as follows, for any random variables $X$ and $R$ over $\mathcal{X}$ and $\mathcal{R}$:

$$\min_{r \in \mathcal{R}} \mathbb{E}[L(A_\theta(X, r))] \leq \mathcal{L}^E(\theta) := \mathbb{E}[L(A_\theta(X, R))] \leq \max_{x \in \mathcal{X}} \mathbb{E}[L(A_\theta(x, R))] \tag{1}$$

More intuitively, given a fixed distribution over our dataset, Yao's Minimax Principle shows:

- 1st inequality: There always exists a deterministic algorithm performing, in expectation over the data, at least as well as the randomized algorithm $A(\cdot, R)$. The process of extracting a particular seed that performs as well as the randomized algorithm is called derandomization. There is however no general efficient recipe for this task.

- 1st and 2nd inequality: Any randomized algorithm will inevitably perform worse on some input than the average performance of the best deterministic algorithm.

At first glance, this principle might seem to nullify any advantage of randomized algorithms over deterministic ones. However, the above principle applies to the setting where we assume a distribution over input data on which we want to perform well on expectation - in particular, when the objective is that of the empirical risk minimization (ERM) on a fixed dataset. Therefore, we recognize that the most common optimization strategy in deep learning is not expected to show benefits from

---

[*]We take the liberty throughout this paper to use the term (transformer) algorithm and model interchangeably.

randomization. In fact, we will show empirically that transformers that are trained on ERM *do not* have random predictions when varying $r$, i.e. ERM does not lead to properly randomized transformers.

### 2.3 RANDOMIZATION CAN BE BENEFICIAL IN ADVERSARIAL SETTINGS.

As discussed, randomness does not provide advantages when considering expected performance on an arbitrary distribution over data. This picture changes drastically when considering adversarial, in particular min-max settings. To understand these settings, we first define a oblivious adversary:

**Definition 1** (Oblivious adversary). *An oblivious adversary possesses complete knowledge of the agent's algorithm but lacks control over the randomness.*

In our context, such an adversary knows the algorithm $A$ and $R$ (the distribution of the seeds) but does not know the outcome of $R$ in advance. However, it can choose the distribution over the input and therefore restrict the input to the most "difficult" instances in $\mathcal{X}$. For example, in rock-paper-scissors it would constantly play paper, if it determines that the opponent deterministically chooses rock. Unlike in other more game-theoretical settings, this adversary isn't a real opponent but rather a theoretical construct: the agent seeks in fact to ensure a favourable outcome irrespective of the input. This setting is closely linked to min-max strategies Wald (1945): an agent facing an oblivious adversary aims to optimize the min-max loss, defined as:

**Definition 2** (Min-max loss). *The min-max objective is the minimization of the following loss:*

$$\mathcal{L}^A(\theta) := \max_{x \in \mathcal{X}} \mathbb{E}[L(A_\theta(x, R))] \tag{2}$$

**Proposition 1.** *[Randomization can be beneficial in worst-case scenarios] Assume that $\mathcal{X}$ is a compact set of $\mathbb{R}^d$ for some d, and that $L$ is continuous. Furthermore assume that there exist a parameter $\theta^*$ and a set of random seeds $(r_i)_{1 \le i \le N} \in \mathcal{R}^N$ such that for each $i$*

$$\max_{x \in \mathcal{X}} L(A_{\theta^*}(x, r_i)) = \min_{\theta, r} \max_{x \in \mathcal{X}} L(A_\theta(x, r)), \text{ and } \bigcap_i \arg\max_{x'} L(A_{\theta^*}(x', r_i)) = \emptyset. \tag{3}$$

*Then there exists a randomized model with strictly smaller loss $\mathcal{L}^A$ than any deterministic model.*

We provide the proof in Appendix A.1. Note that the above subsumes the more straightforward argument that no randomization is expected if the model can fit the data perfectly. Intuitively, the proposition shows that whenever there are several possible optimal functions for $\mathcal{L}^A$ that can be implemented within our model class, such that there is no input $x$ which is adversarial to all of them, then it is more beneficial to have a distribution over such functions than to deterministically encode a single one.

As argued by Rice et al. (2021), in the context of robust optimization it is often advantageous to relax the strict adversary of equation 2 by optimizing the $q-$norm of the expected loss of the model distribution for $q > 1$, i.e.

$$\min_\theta \mathcal{L}^q(\theta) = \min_\theta \mathbb{E}[|\mathbb{E}[L(A_\theta(X, R))|X]|^q]^{1/q} \tag{4}$$

Here, the conditional expectation $\mathbb{E}[L(A_\theta(X, R))|X]$ is the averaged loss over $R$. The outer expectation is over inputs. Note that with $q = 1$, $\mathcal{L}^q = \mathcal{L}^E$ and with $q = \infty$ we obtain $\mathcal{L}^q = \mathcal{L}^A$.

### 2.4 OUR TRAINING OBJECTIVE.

Based on the previous description, summarized in Proposition 1, of when randomization is beneficial, we propose the following practical training objective:

$$\arg\min_\theta \hat{\mathcal{L}}^q(\theta) = \arg\min_\theta \left( \frac{1}{n} \sum_{i=1}^n \left( \frac{1}{m} \sum_{j=1}^m L(A_\theta(x_i, r_j)) \right)^q \right)^{1/q}. \tag{5}$$

Note that this a biased approximation of equation 4. Although approximating the $q$-norm by such Monte Carlo sampling in practice is known to slowly converge to the expected value, see Rice et al. (2021), we stick to it here now and leave further investigations of refined sampling to future work.

In our training objective, we introduce, compared to the common stochastic approximation of the expected loss over the data, two new hyperparameters namely $m$, the number of random seeds we consider, and $q$. We will analyze the role of these in our experimental results section, focussing on the sensitivity of $q$, as it shifts between ERM and adversarial training.

We stress that finding adversarial examples to compute the loss in equation 2 is difficult, especially in settings where computing gradients poses challenges such as in language with discrete inputs or in reinforcement learning (RL). Therefore, the relaxation of $q < \infty$ approximating the min-max adversary leads to a practical loss for which no explicit adversary is computed. Finally, note that the adversarial loss upper bounds the expected loss, i.e., $\mathcal{L}^A(\theta) \geq \mathcal{L}^E(\theta)$. Nevertheless it is not expected that optimizing $\mathcal{L}^A(\theta)$ will lead to better $\mathcal{L}^E(\theta)$ compared to when optimizing $\mathcal{L}^E(\theta)$ directly.

In the next section we show how we can turn the above theoretical considerations into practice and present experimental results where optimization leads to randomized algorithms in transformers.

## 3 EXPERIMENTAL RESULTS

Before we present experimental results, we first describe an evaluation protocol to validate experimentally that

1. the algorithm implemented by a transformer model after optimizing on $\mathcal{L}^q$, for certain $q$ and $m$, induces randomization: The distribution of the random variable $A_\theta(x, R)$ does not collapse i.e. is not degenerate. Therefore $r \to A_\theta(x, r)$ is not constant.

2. the randomized transformer algorithm found is performing, compared to baselines described below, better on data that is adversarially chosen.

3. training on expected risk $\mathcal{L}^E$ will not lead to randomization i.e. now $r \to A_\theta(x, r)$ is constant and does not possess the associated robustness against adversaries.

Finally, we investigate the robustness and scaling of proper randomization with respect to the novel hyperparameters $m$ and $q$. In order to provide the results in a structured way, we present an evaluation protocol which we follow when discussing our empirical findings in the following.

### 3.1 EVALUATION PROTOCOL AND BASELINES

In most of our experiments, we compare the following four transformer models:

1. $A_{r_0}^E$ - **trained on expected loss, single seed**: This is the transformer baseline, trained on ERM, which is by definition deterministic. The transformer is $x \mapsto A_\theta(x, r_0)$ where $r_0$ is a static seed selected at the beginning of training from $\mathcal{R}$. The same $r_0$ is used for evaluation. The transformer is thus trained on $\mathcal{L}_{r_0}^E(\theta) = \mathbb{E}[L(A_\theta(X, r_0))]$.

2. $A_{r_0}^q$ - **trained on relaxed adversarial loss, single seed**: This is the same transformer as above, but trained on the (relaxed) adversarial loss $\mathcal{L}_{r_0}^q(\theta) = \mathbb{E}[L(A_\theta(X, r_0))^q]^{1/q}$ for some parameter $q$. In practice, we approximate the objective using equation 5, where $r_i = r_0$ for all $i$. This transformer is closely related to robust optimization e.g. distributionally robust optimization (DRO), see related works section.

3. $A_R^E$ - **trained on expected loss, multi seed**: This transformer model is given as input a random seed $r$ which is sampled from $\mathcal{R}$, but trained on ERM. Concretely, the transformer is trained on $\mathcal{L}_R^E(\theta) = \mathbb{E}[L(A_\theta(X, R))]$.

4. $A_R^q$ - **trained on relaxed adversarial loss, multi seed**: This is our transformer model of main interest trained on our proposed objective in equation 5. This is the model we expect to randomize itself and the focus of our investigation.

Given these models, we evaluate them in the following ways:

1. For $A_{r_0}^E$ and $A_{r_0}^q$, the loss $L(A_\theta(x, r_0))$ for an input $x$ is computed by using the same seed $r_0$ which was used for training. In all our experiments, $A_\theta(x, r_0)$ parametrizes a discrete decision to take, such as the recalling of bits in the associative recall task (Section 3.2), the coloring of vertices in the graph coloring task (Section 3.3), or the action to take in

the grid world task (Section 3.4). In all these cases, $A_\theta(x, r_0)$ produces a $D$-dimensional vector where $D$ is the number of possible discrete decisions. We may train the model by interpreting the prediction as parametrizing e.g. a categorical distribution. To compute the success percentage however, we record whether the argmax of the prediction $A_\theta(x, r_0)$ is correct or not. Furthermore, we also report the success percentage by sampling from the distribution parametrized by $A(x, r_0)$, in the case the loss $L$ allows for such interpretation (e.g. $L$ is the cross entropy loss). We denote these by $\boldsymbol{A_{r_0}^E}$-**sampled** and $\boldsymbol{A_{r_0}^q}$-**sampled**, resp.

2. For $A_R^E$ and $A_R^q$, the loss for an input $x$ is computed as the expected loss over random seeds, i.e. $\mathbb{E}[L(A_\theta(x, R))]$. For the success percentage given an input, we record the fraction of seeds $r$ for which the argmax of the prediction $A_\theta(x, r)$ is correct. Furthermore, we report the success rate of majority voting of the transformer predictions across seeds, which we denote by $\boldsymbol{A_R^E}$-**majority** and $\boldsymbol{A_R^q}$-**majority**, resp. See Appendix E.1.1 for more details.

Finally, we report the performance of these models when averaging over the input distribution (Average) and measure the performance on *adversarial* inputs by reporting the 95th percentile performance values over the input distribution ($95^{th}$-percentile). We use the common transformer architecture and variations throughout our experiments; more details in Appendix A.2.

## 3.2 RANDOMIZED TRANSFORMERS SOLVE ASSOCIATIVE RECALL

We start by revisiting the associative recall task which we introduced in the introduction, cf. Figure 1. Here we train transformers with linear self-attention layers i.e. we replace the standard softmax operation with the identity function $E \leftarrow E + (QK^T \odot M)VW_P$. This architecture change allows the transformer to be regarded as a *fast-weight* programmer Schmidhuber (1992); Schlag et al. (2021) where an internal fixed size memory matrix can be overwritten with incoming information. Studying memory allocation issues with transformer variants has seen considerate interest recently (Gu & Dao, 2023; De et al., 2024; Orvieto et al., 2023; Arora et al., 2023; von Oswald et al., 2023b; Zucchet et al., 2024). We give more details on the tokenization and how we provide randomness in Appendix B. The transformer is provided with the inputs that are the to-be-remembered key and value pairs i.e. $E = [e_0, \ldots, e_N]$ with $e_i = [v_i, k_i, \text{seed}]$, where $v_i$ is a binary vector of $d$ bits $v_i = [v_{i0}, \ldots, v_{id}]$ with $v_{ij} \in \{0, 1\}$ and $e_N = [\cdot, k_i, \text{seed}]$. We train the model on the sum of binary cross-entropy (CE) between ground truth bits and transformer predictions: $L(A_\theta(E, r), v_i) = \sum_j \text{CE}(\hat{v}_j, v_{ij})$. We report

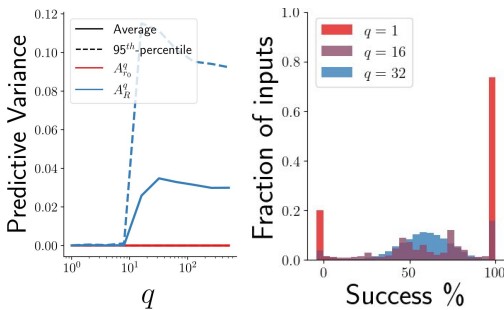

Figure 2: *Left*: Variance of the predictive probability conditioned on the input sequence, w.r.t. increasing $q$: Larger $q$ leads to randomized transformer models with non-zero output variance over the seeds, with a phase transition around $q \approx 16$. *Right*: Histogram showing the fraction of inputs, all of length $N = 20$, with varying recall success rates, when training $A_R^q$ with various $q$. For ERM training i.e. $q = 1$, we see the transformer producing essentially binary predictions, i.e.the predictions, over seeds r, are either correct or incorrect. For $q > 1$, we see randomization emerging, with non-zero success rate on all input for $q = 32$.

the performance of the different models, including when varying $q$ and $m$, in Figure 2 and 3.

We start by analyzing ERM trained transformers for which we do not expect randomization to emerge, i.e. $A_R^E = A_R^q$, $A_{r_0}^E = A_{r_0}^q$ with $q = 1$, and their variants, Figure 3A. All models perform similarly, concluding that ERM models are *not* benefiting from additional randomness, even when random seeds are provided. Furthermore, even transformers that could potentially leverage randomness, i.e. $A_R^E$, collapse in their predictions, see Figure 2 where we show the collapsed predictive variance induced when varying seeds for $A_R^E$. This behaviour changes when increasing $q$. Indeed, when choosing for example $q = 100$, we observe that randomization emerges demonstrated by increased predictive variance and gradual, instead of bi-modal, success rate, see Figure 2. Performance wise, the randomized $A_R^q$ (blue) improves significantly on worst-case inputs and especially $A_R^q$-majority (green), performs almost optimally on all inputs, see Figure 3B. We denote $A_{r_0}^E$ as "deterministic" in Figure 1 where we show for illustrative purposes the performance of the 95th percentile.

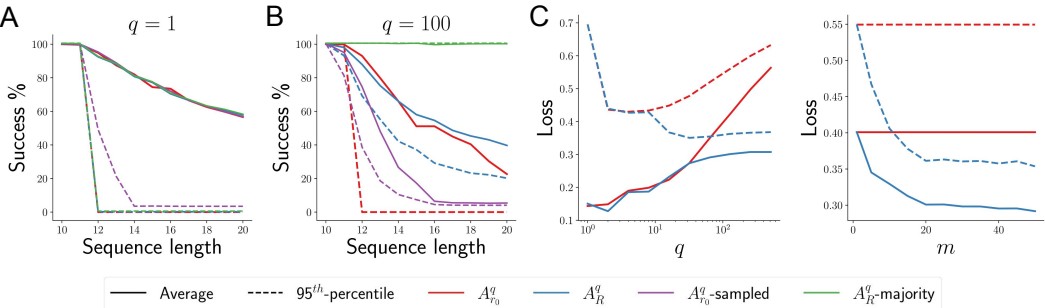

Figure 3: **Associative recall analyses**. **A**): Performance of models trained on ERM does not improve when provided with additional seeds, cf. overlapping lines of $A_{r_0}^q$ and $A_R^q$. **B**): Models trained with our objective ($q = 100$) do exhibit drastic improvements, especially with majority voting, compared to models trained with a single fixed seed in their input. **C**): Influence when training with different $q$ and $m$, measured on the loss with $q = 1$. First, models show gradual improvement when increasing $q$, with randomness emerging over a certain threshold. Second, when fixing $q = 100$, outer right plot, we observe already for small $m > 1$ improvements over the deterministic counterpart.

We conclude that training linear self-attention transformers to solve associative recall tasks instills a randomized strategy inside the transformer which outperforms deterministic counterparts.

### 3.3 RANDOMIZED TRANSFORMERS CAN SOLVE GRAPH COLORING PROBLEMS

Randomization has numerous applications in combinatorics as a theoretical and practical tool to design powerful and strikingly simple algorithms. One prominent use case of randomized algorithms is solving graph coloring problems, with the aim to color the vertices of a graph $G = (V, E)$ in such a way that vertices connected by edges are colored differently. Randomized algorithm are particularly useful in distributed settings in which each vertex can only see and talk to its neighbors. See Appendix E.4 for a detailed description. We consider the problem of 3-coloring cycles, see Figure 4. While the problem per se is trivial, the goal in the distributed setting is to minimize the time/number of communication rounds until the coloring is found.

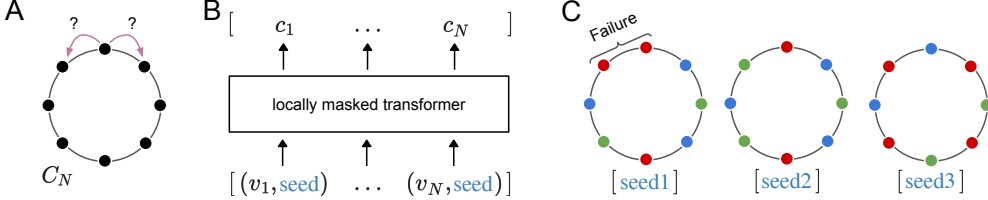

Figure 4: **Transformers solving graph coloring problems**. **A**): We study distributed vertex coloring problems on cycles $C_n$ where every vertex can only communicate with its immediate neighbors. **B**): A locally masked transformer, which can only attend to the immediate neighboring vertices on the graph by an appropriate attention mask, has to decide on the vertex color. **C**): If the transformer realizes a fixed mapping of vertex id $v_i$ to a color, an adversary can provide an input permutation for which the transformer fails to generate a correct coloring. However, when trained with our objective, the transformer model, to protect itself against this adversary, implements a randomized strategy. The coloring computed by the transformer for a fixed graph, now depends on the random seed and will fail in some and be correct in others. With the probability of being correct hopefully being large.

We consider $N = 10$ vertices, each with a unique id from 1 to $N$. Each task consists of the 3-coloring problem i.e. a cycle drawn uniformly from the set of possible cycles with $N$ vertices. A transformer input are $N$ tokens, each consisting in a distinct $N$-dimensional 1-hot vector, representing the $N$ vertices. The random noise is concatenated to each of these tokens, see Figure 4. The adjacency matrix of the cycle is translated into an attention mask of the transformer, such that each vertex can only attend to its neighbors on the cycle. No other information about the particular configuration of the cycle is given. Every token stream then outputs the distribution over its color. Details in App. C.

The objective is a partial coloring loss, which is an upper bound of the probability of the model outputting an invalid coloring of the graph. Specifically, given all edges $E$ of a given cycle, the loss is defined as: $\sum_{(u,v) \in E} \sum_{c \in \{1,2,3\}} \mathbb{P}(C(u) = c)\mathbb{P}(C(v) = c)$ which upper bounds the quantity

$\mathbb{P}(\bigcup_{(u,v)\in E} C(u) = C(v))$ where $C(u)$ is the random variable assigning a color to $u$, sampled from

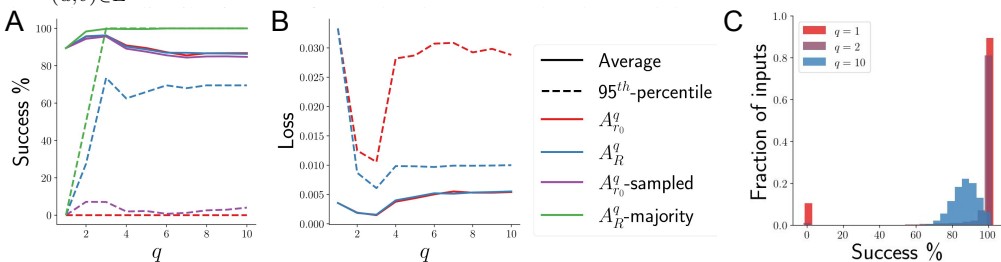

Figure 5: **Graph coloring performance analyses**. **A)**: Performance of models trained on $\mathcal{L}^q$ for varying value of $q$. As $q$ increases, $A_R^q$ learns to leverage randomness to implement a randomized algorithm, inducing large improvements of worst-case performance compared to the deterministic counterpart $A_{r_0}^q$. Furthermore, majority voting boosts performance to close to optimal performance (green). Both the average and percentile curves are computed over all possible cycles of size N=10. **B)**: Influence of varying $q$ during training evaluated on the loss with $q = 1$. After training with $q > 3$ a clear advantage of the randomized model is observed. **C)**: Histogram over all possible inputs, of the transformers success probability when varying $q$ when training $A_R^q$. When increasing $q$ we observe non-zero success rate due to randomization.

For our experiments we choose the transformer depth and the task difficulty defined by $N$ such that there cannot be enough communication between vertices to coordinate and return a valid coloring. A deterministic strategy may work on a large fraction of possible cycles, but may fail on the remainder. To investigate the effect of the adversarial loss on the emergence of proper randomness, we again train $A_{r_0}^q$ and $A_R^q$ for varying adversarial strength $q$. We present our results in Figure 5. We see that at $q = 1$ the performances of the various transformers are almost identical. In particular, despite having a random source at its disposition, both $A_R^q$ and $A_R^E$ learn a degenerate transformer that is, for each input, either correct or incorrect for all seeds, cf plot on the right, $q = 1$. As $q$ increases proper randomness emerges and, in particular, the performance of $A_R^q$-majority improves significantly close to $100\%$ success probability. This signals that the transformer learned a randomized algorithm, where each seed underfits different parts of the input space such that, on expectation, the transformer returns a correct output for each input with relatively high probability. Intriguingly, $A_{r_0}^q$-sampled, despite leveraging randomness at the output, does not recover the same performance as the properly randomized transformer, even though the objective optimizes the predictive distribution. To conclude, we are able to show that powerful randomization emerges within transformers from first principles due to optimization in the graph coloring setting as well.

## 3.4 RANDOMIZED TRANSFORMER AGENTS EXPLORE GRID WORLDS

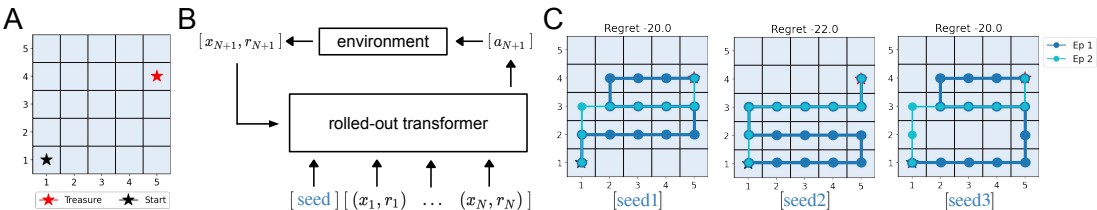

Figure 6: **Transformer agents exploring grid worlds in-context**. **A)**: The transformer agent is trained to explore a simple grid efficiently and search for a randomly placed treasure. **B)**: The transformer agent's output provides the next action (right, left, up, down) for which the environment returns the next state and a reward. The tuple is appended to the transformer model's context. **C)**: Given different initial seeds, the agent chooses different explorations as it is adversarially trained to avoid a deterministic strategy which could be easily attacked leading to high negative regret. We observe that after an initial exploration phase (Ep 1), the transformer exploits in a second episode (Ep 2) the previously obtained knowledge i.e. the treasure's position and approaches it optimally.

To showcase that our objective can discover randomized algorithms when lacking differentiability, we study rolled out transformers which we train to explore and exploit simple grid worlds with evolutionary strategies, see Appendix D for details on the environment dynamics as well as training

setup. We stress that we do not consider reinforcement learning strategies e.g. based on policy gradients, as this requires sampling from a policy and therefore uses additional randomness. Here we aim to isolate the effect of randomness provided to the input of the transformer i.e. the first token.

As illustrated in Figure 6 C, after training the transformer on our objective, the model successfully implements a randomized strategy: Over different random seeds, the transformer explores the grid differently in the first episode while exploiting in the second episode its knowledge about the treasure location. The performance during training when comparing to $A_R^E$ is shown in Figure 7 where we see again $A_R^q$ outperforming wrt. worst-case inputs and coming close to $A_R^E$ on average. In Appendix D, we also provide the other baselines $A_{r_0}^q, A_{r_0}^E$ where the former did not properly train given our hyperparameter scan, and where we observe that $A_{r_0}^E \approx A_R^E$, see Figure 10. To quantify randomization, we visualize the average time the transformer agent visits a cell for the first time - showing its randomized exploration when considering $A_R^q$ instead of $A_R^E$, see Figure 7.

To conclude, even in this non-differentiable setting, optimization forces the transformer to use the provided randomness and instils efficient randomized exploration inside the transformer weights.

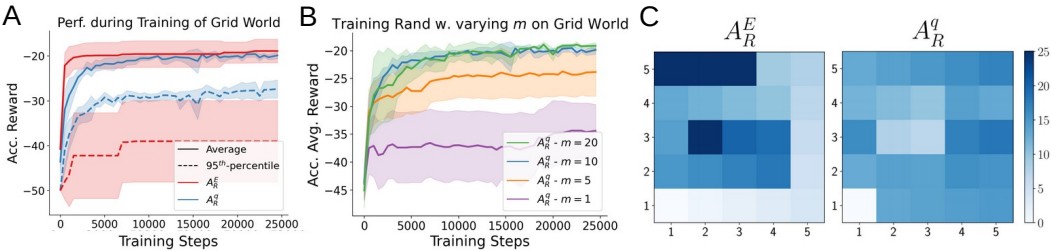

Figure 7: **Grid world performance analyses**. **A)**: Accumulated reward over training transformers on our adversarial loss leading to models $A_R^E$ and $A_R^q$. The randomized model, while competitive on average, drastically outperforms on adversarial inputs. **B)**: The adversarial loss requires several seeds for proper optimization. We scan $m$ for the transformer $A_R^q$ and observe that with $m \geq 10$ we reach close to optimal average performance, while being robust to adversarial inputs. **C)**: Average timestep at which the agent visits a given cell for the first time capped at 25. Despite having access to random seed, $A_R^E$ degenerates to a deterministic trajectory. On the other hand, $A_R^q$ learns a collection of trajectories such that on average, the timestep of the first visit is relatively uniform over cells.

## 4    DISCUSSION & RELATED WORK

**Summary & Limitations.** Randomness is an integral part of computer science. In particular, there is a long history in developing and analyzing randomized algorithms. In this paper, we build a bridge between this intriguing class of algorithms and deep learning and study under which circumstances optimization learns powerful *randomized transformer algorithms*. By optimizing a well-motivated objective we show that randomness is chosen by optimization to endow transformer models with remarkable properties similar to those which classic randomized algorithms posses. Specifically, our learned randomized transformer algorithms significantly outperform their deterministic counterparts on adversarial inputs. Furthermore, simple majority voting among predictions computed on different seeds can boost performance far beyond deterministic transformers. Nevertheless, the provided evidence is only a first conceptual step into how to learn powerful randomized neural network algorithms, although we believe that the presented theory is general and applies to more practically relevant and large scale settings. Furthermore, it is noteworthy that the computational intensity of our approach limits the applicability and scalability of the current form of our approach. Note that when comparing to ERM trained models, the hyperparameter $m$ scales the memory as well as train and inference time (roughly) by a factor of $m$. We leave scaling up our approach for future work.

**Sampling** from distributions, such as policies or probabilistic models, requires randomness. However, it remains unclear and a key research direction to analyze the differences, advantages, and synergies of sampling from policies or (Bayesian) probabilistic models versus input-level sampling, as proposed here. In deep learning, methods like Markov-Chain-Monte-Carlo (MCMC) and variational inference (Kingma & Welling, 2013; Rezende et al., 2014), loosely including dropout (Srivastava et al., 2014; Gal & Ghahramani, 2015) and ensembling (Lakshminarayanan et al., 2017), are related to

our approach. This also applies to generative adversarial networks (GANs) (Goodfellow et al., 2014). These methods use randomness to sample from prior or (approximate) posterior distributions. However, we emphasize that in these methods, randomness is an intentional design choice and forced upon the algorithms computational graph rather than a learned characteristic. Therefore, when trained with our objective, weight distributions that randomize network predictions might be learned without a Bayesian formalism. This suggests a connection between Bayesian inference and our method leading to randomized neural networks based on, for example, "adversarial weight uncertainty".

**Relationship to neuroscience.** Randomness is also hypothesized to be useful for the brain. On the one hand, it is well-known that the brain is exposed to noise but also produces activity that resembles chaos (London et al., 2010; Srajer et al., 1996; Faisal et al., 2008). On the other hand theories on functional properties due to criticality and learning algorithms that harness the randomness within the activity of recurrent neural networks exist (Lengler et al., 2013; Moss et al., 2004; Shew & Plenz, 2012; Jaeger & Haas, 2004; Sussillo & Abbott, 2009). Nevertheless, an objective that leads, in theory and in practice, to learning and leveraging randomness is not known to us. Here, we provide such an objective: the desire to perform well in worst-case scenarios. Furthermore, we provide evidence that optimization of the objective actually leads to randomized behaviour. The presented results provide an explanation of human and animal behavior and decision-making processes which resemble taking actions leveraging randomization (Maoz et al., 2019; Glimcher, 2005; Noble & Noble, 2018; Domenici et al., 2008; Braun, 2021), hypothesizing that learning in the brain or evolution might optimize for some relaxation of worst-case behaviour as well.

**Meta-learning learning transformer algorithms.** Through the advent of the transformer and large foundation models we are diverging from the classic perspective of neural networks learning abstract complex data representation LeCun et al. (2015). Indeed, large language models can be regarded as flexible tools and algorithms Weiss et al. (2021); Lindner et al. (2023); Giannou et al. (2023b) which can learn in-context Brown et al. (2020); Garg et al. (2022); Akyürek et al. (2023); von Oswald et al. (2023a); Giannou et al. (2023a); Liu et al. (2023a); Li et al. (2023) and be re-configured by prompting Wei et al. (2022); Radford et al. (2019); Lester et al. (2021). We expand the class of algorithms that transformers can learn from data, to the family of randomized algorithms. We hope that through this work, we put more emphasis on the neural algorithms perspective with exciting future research directions aiming towards distilling known, e.g. stochastic gradient descent, or discovering novel, unknown randomized algorithms hidden inside the trained weights of deep neural networks.

**Robustness.** At the heart of our objective lies the desire to perform better in adversarial settings, a goal shared by established deep learning and RL methods. This goal is closely related to having more robust models. In this context, the incorporation of randomness has already proven to be beneficial in many domains. For instance, adversarial bandits have been extensively studied within the broader field of RL, particularly as a robust extension of the classical multi-armed bandits. Foundational work in adversarial bandits by Auer et al. (2002) introduces the EXP3 algorithm which uses randomization to cope with oblivious adversaries. This line of research is already conceptually close to our grid world task, where smart exploration is learned to protect the agent against adversarial environments. Due to these connections, we speculate that our approach could allow for randomized algorithms being learned when trained with our objective in the domain of adversarial bandits and RL.

In another line of work, in the domain of supervised learning, common adversarial training techniques are entirely deterministic. Some of them (Madry et al., 2018) employ gradient-based attacks to augment training data with the most challenging perturbations, aiming to ensure consistent output within the neighborhood of any input. Other work investigated objectives aiming at prepare a model to perform well under distributional shifts (Arjovsky et al., 2020). These objectives are very similar to our training objective (Duchi & Namkoong, 2020; Rahimian & Mehrotra, 2022), except for the random component. Recent advancements have focused on improving robustness through randomization, either of the model parameters or of the inputs, during both training and inference Rakin et al. (2018). These methods, when combined with adversarial training, are closely related to our proposed randomization technique. However, our approach determines the adversarial input based on the expected loss over random seeds, rather than relying on a single sampled seed i.e. $m = 1$ - a crucial distinction to our objective which relied on $m > 1$ as shown by our experiments.

Randomization is also used in randomized smoothing Cohen et al. (2019), a certified adversarial robustness method. This technique computes the majority vote over random perturbations of the input rather than over random seeds for a given input. We hypothesize that our procedure may offer

better robustness guarantees by additionally randomizing decision boundaries. Overall, our technique is readily applicable to methods aimed at defending against adversarial attacks and distribution shifts Sinha et al. (2020); Zou et al. (2023), positioning it as a promising avenue for future research.

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

# A APPENDIX

We provide here additional results and training details to reproduce the experiments one by one, as well as a proof of proposition 1.

We give first a short overview of the compute budget used for this project. For the associative recall task as well as the graph coloring problem, we estimate a total compute budget using 4 Nvidia RTX 4090 for a month. For the grid world problem, we use a cluster of 16xA100 GPUs which we used to scan over the hyperparameters on and off over a total of 2 weeks.

## A.1 PROOF OF PROPOSITION 1

We first restate Proposition 1:

**Proposition 1** (Randomization can be beneficial in worst-case scenarios). *Assume that $\mathcal{X}$ is a compact set of $\mathbb{R}^d$ for some d, and that L is continuous. Furthermore assume that there exist a parameter $\theta^*$ and a set of random seeds $(r_i)_{1 \le i \le N} \in \mathcal{R}^N$ such that for each i*

$$\max_{x \in \mathcal{X}} L(A_{\theta^*}(x, r_i)) = \min_{\theta, r} \max_{x \in \mathcal{X}} L(A_\theta(x, r)), \text{ and } \bigcap_i \arg\max_{x'} L(A_{\theta^*}(x', r_i)) = \emptyset. \quad (6)$$

*Then there exists a randomized model which has a strictly smaller loss $\mathcal{L}^A$ than any deterministic model.*

*Proof.* Assume there exist a parameter $\theta^*$ and a set of random seeds $(r_i)_{1 \le i \le N} \in \mathcal{R}^N$ such that for each $i$ equation 6 holds.

Let $M = \min_{\theta, r} \max_{x \in \mathcal{X}} L(A_\theta(x, r))$. Any deterministic model, i.e. any choice of parameter $\theta$, and fixed seed $r \in \mathcal{R}$, will yield an adversarial loss of at least $M$. On the other hand, consider $R$ following a uniform distribution on the $(r_i)_i$.

Let us now assume that

$$\inf_{x \in \mathcal{X}} \mathbb{E}[L(A_{\theta^*}(x, R))] = \inf_{x \in \mathcal{X}} \frac{1}{N} \sum_i L(A_{\theta^*}(x, r_i)) = M$$

We can then fix a sequence $(x_n)_{n \in \mathbb{N}} \in \mathcal{X}^{\mathbb{N}}$ such that $\lim_{n \to \infty} \mathbb{E}[L(A_{\theta^*}(x_n, R))] = M$. Since $\mathcal{X}$ is a compact set, without loss of generality we can assume $(x_n)_{n \in \mathbb{N}}$ to converge to some $x^*$. By continuity of $L$, we have $\mathbb{E}[L(A_{\theta^*}(x^*, R))] = M$. Thus, necessarily, for all $i$, $L(A_{\theta^*}(x^*, r_i)) = M$, i.e. $x^* \in \bigcap_i \arg\max_{x'} L(A_{\theta^*}(x', r_i))$ which contradicts that assumption in equation 6.

We therefore have, for each $x \in \mathcal{X}$,

$$\mathcal{L}^A(\theta^*) = \inf_{x \in \mathcal{X}} \mathbb{E}[L(A_{\theta^*}(x, R))] < M$$

$\square$

## A.2 THE TRANSFORMER ARCHITECTURE AND THE RANDOM SEED ENCODING

We provide here a short recap of the transformer neural network architecture (Vaswani et al., 2017; Phuong & Hutter, 2022) that we use throughout our experiments. We define a transformer of depth $K$ as being the transformation in blocks of the following two operations $K$ times. First, we are considering input tokens $E \in \mathbb{R}^{L \times d_m}$ for a sequence of length $L$, to which we add common positional encodings $P \in \mathbb{R}^{L \times d_m}$, and either add or, as we do in this paper in most experiments, concatenate the *random seed encoding* (RSE). This is simply $R \in \mathbb{R}^{L \times d_r}$ or $R \in [0,1]^{L \times d_r}$ where each element of $R$ is sampled randomly from a unit interval or a Normal distribution. Then, a self-attention layer followed by a multi-layer-perceptron transform the tokens by first computing queries, keys and values $Q, K, V = EW_q, EW_k, EW_v$ with which we then update $E$ as

$$E \leftarrow E + (\text{softmax}(QK^T) \odot M)VW_P \quad (7)$$
$$E \leftarrow E + \sigma(EW_1)W_2 \quad (8)$$

where $W_q, W_k, W_v \in \mathbb{R}^{d_m \times d_k}$ and $W_p \in \mathbb{R}^{d_k \times d_m}$ as well as $W_1 \in \mathbb{R}^{d_m \times 4d_m}, W_2 \in \mathbb{R}^{4d_m \times d_m}$ are learnable parameter matrices. The softmax operation is applied row-wise. $M$ is a 0-1 mask that controls the attention span, $\sigma$ a non-linearity. We apply LayerNorm Ba et al. (2016) to the inputs of the self-attention layer as well as the MLPs. We now present results where we trained the just described transformer, as well as variants, on our training objective. We stress that these models are parametrized in a way, although potentially only by taking positional encodings into account, which allows to condition computation on the input position and therefore ignore the parts where we provide randomness. We see that this is indeed the case when training on ERM and show this in the following.

## B  RANDOMIZED TRANSFORMERS SOLVE ASSOCIATIVE RECALL TASKS

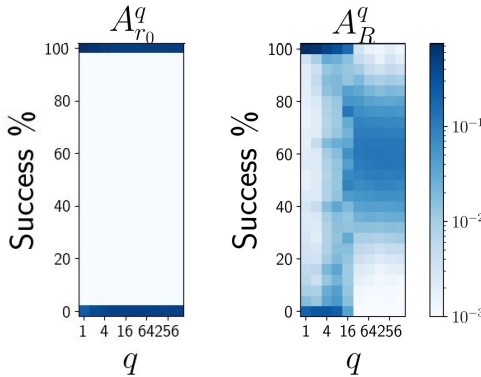

Figure 8: Fine-grained performance analyses of $A_R^q$ models trained with varying $q$ on the associative recall task. When increasing $q$, above ERM (q=1), we observe non-zero success rate on all inputs due to randomization. This success probability, see main text, can be increased by majority voting.

Table 1: Hyperparameters for the associative recall task.

| Hyperparameter | Value |
|---|---|
| Dataset | Randomly generated binary value vectors with $d = 5$ and corresponding one-hot encodings as the keys |
| Tokenization & RSE | One token is the concatenated vector $[v_i, k_i, r_i]$ where $r_i$ is (the same) random binary vector for all $i$ with $d_r = 10$. |
| Context size | Variable size from 8 - 20, see Figure 1 |
| Optimizer | Adam (Kingma & Ba, 2015) with $\epsilon = 1e^{-5}, \beta_1 = 0.9, \beta_2 = 0.95$ |
| Hyperparamters of our objective | $m = 30$ and $q = 100$ |
| Batchsize | 512 |
| Gradient clipping | Global norm of 1. |
| Positional encodings | We add standard positional encodings. |
| Architecture details | 2 transformer blocks with linear self-attention, 1 head, key size 5, token size 15, no input- but output-embedding |
| Attention mask details | Causal mask |
| Weight init | Truncated normal initial with variance computed by common fan-in, bias parameter to zero. We scale all weight matrices before a skip connection with $\frac{1}{2\sqrt{N}}$ with $N$ the number of layers. |
| Learning rate scheduler | Linear warm-up starting from 0 to 0.003 for 2000 steps annealed to 0.0003 |
| Standard deviation / Stat. robustness | We average all results over 5 random seeds. We omit showing the deviation due to negligible differences across seeds. |

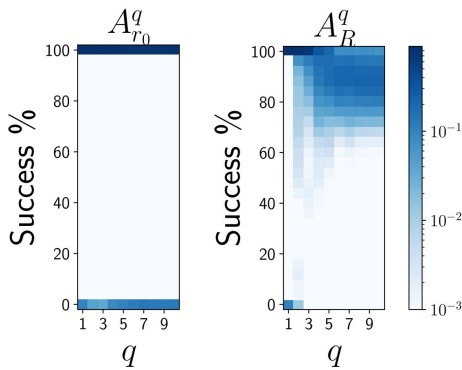

Figure 9: Fine-grained performance analyses of $A_R^q$ models trained with varying $q$ on the 3-coloring problem. When increasing $q$, above ERM (q=1), we observe non-zero success rate on all inputs due to randomization. This success probability, see main text, can be increased by majority voting.

## C  RANDOMIZED TRANSFORMERS SOLVE A GRAPH COLORING TASK

Table 2: Hyperparameters for the graph coloring task.

| Hyperparameter | Value |
|---|---|
| Dataset | Random permutation of graph indices |
| Randomness | Single random floating point concatenated to embedding |
| Tokenization | Every token is the concatenated vector $[v_i, k_i, r_i]$ where $r_i \in [0, 1]$ and $d_r = 1$. |
| Context size | 10 i.e. we only consider cycles $C_{10}$ |
| Optimizer | Adam (Kingma & Ba, 2015) with $\epsilon = 1e^{-3}$, $\beta_1 = 0.9$, $\beta_2 = 0.95$ with weight decay of 0.1 |
| Training steps | 300000 |
| Hyperparamters of our objective | $m = 10$ and $q = 10$ |
| Batchsize | 256 |
| Gradient clipping | Global norm of 1. |
| Positional encodings | We add standard positional encodings. |
| Architecture details | 2 blocks of self-attention, 1 head, key size 16, token size 16, no input- but output-embedding |
| Attention mask details | Mask that only allows to observe direct neighboring tokens |
| Weight init | Truncated normal initial with variance computed by common fan-in, bias parameter to zero. We scale all weight matrices before a skip connection with $\frac{1}{2\sqrt{N}}$ with $N$ the number of layers. |
| Learning rate scheduler | Linear warm-up starting from 0 to 0.001 for 1000 steps annealed to 0.0001 |
| Standard deviation / Stat. robustness | We average all results over 5 random seeds. We omit to show the deviation due to negligible differences across seeds. |

## D  TRAINING TRANSFORMER AGENTS TO EXPLORE AND EXPLOIT GRID WORLDS.

We provide here additional results and details accompanying the main text.

Here, we first described the environment dynamics, loss functions and further details. For every task, we set the starting point of the agent in the lower left corner of a $5 \times 5$ grid but sample a random treasure location. The aim of the agent is now to explore the grid world efficiently. After 25 steps, the agent's position is reset and is given an additional 25 steps to exploit the (potential) knowledge of the treasure location. After every step, the transformer agent is either given a reward of 1 if the treasure is found and $-0.1$ otherwise, i.e. $L(A_\theta) = \sum_i r_i$. To have the ability to explore differently,

the first token of the model is a vector sampled from a normal distribution. Given this vector, the transformer first emits an output probability over the 4 actions {left, up, right, down} from which we extract the action by computing the argmax. Given the environment response providing the reward and next state, the concatenated embedding of the $(i, j)$ coordinates as well as the reward is appended to the sequence which is fed back to the same transformer agent. If the transformer finds the treasure, the episode terminates. To compute an adversary in this setting, we do not fall back to the $q$-norm relaxation. Rather we compute the worst-case treasure position for the current transformer strategy by brute force iteration over all treasure positions. The adversarial position in the grid is the one with the lowest accumulated reward. We stress the scalability issues, which we leave for future work, of this strategy to compute an adversary for most practical environments. To optimize the transformer weights, we use parameter-exploring policy gradients (PEPG) (Sehnke et al., 2010) a common gradient-free optimization method. PEPG estimates gradients based on evaluating populations of weight samples. Unlike methods like OpenAI-ES (Salimans et al., 2017), it leverages a diagonal search covariance to infer directions of improvement.

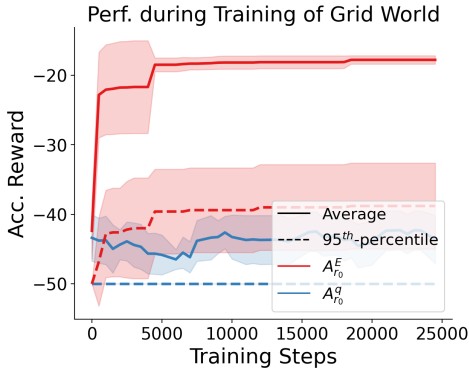

Figure 10: Performance of models $A_{r_0}^E$ and $A_{r_0}^q$. While $A_{r_0}^q$ is difficult to train, given the hyperparameter sweep considered, $A_{r_0}^E$ performs similarly as $A_R^E$, see main text.

Table 3: Hyperparameters for the grid world task.

| Hyperparameter | Value |
|---|---|
| Dataset | Randomly generated treasure location in $5 \times 5$ grid |
| Randomness | Diverging from the other setups, here we only provide a single first random token i.e. $E[0]$ where every element is a sample from a Normal distribution. |
| Tokenization | One token is the concatenated vector $[\text{emb}(i), \text{emb}(j), r]$ where emb is a learnable embedding vector and $(i, j)$ the current position of the token. $r \in \{-0.1, 1\}$ is the reward at that time step. |
| Context size | $2 \times 25 + 1$ for two episodes of 25 steps each plus one initial random vector |
| Optimizer | PGPE with a population size of 300, center lr = 0.01, std lr = 0.01 and init std = 0.03 We swept over the following parameters and choose the best for the randomized transformer as well as deterministic baseline center lr $\in \{0.03, 0.01, 0.003, 0.001\}$ , std lr $\in \{0.03, 0.01, 0.003, 0.001\}$ and init std $\in \{0.03, 0.01, 0.003, 0.001\}$. |
| Hyperparamters of our objective | $m = 10$ and $q = \infty$ |
| Positional encodings | We add standard positional encodings. |
| Architecture details | 2 transformer blocks, 4 heads, key size 10, token size 80, we use an output-embedding |
| Attention mask details | Causal mask |
| Weight init | Truncated normal initial with variance computed by common fan-in, bias parameter to zero. We scale all weight matrices before a skip connection with $\frac{1}{2\sqrt{N}}$ with $N$ the number of layers. |

## E   SOME BACKGROUND ON RANDOMIZED ALGORITHMS

As we motivated in the introduction, we are aiming in this paper to learn randomized algorithms within neural networks, in particular transformers. Here we want to provide a short background on randomized algorithms and their features.

A randomized algorithm is simply  *" ... one that receives, in addition to its input data, a stream of random bits that it can use for the purpose of making random choices. Even for a fixed input, different runs of a randomized algorithm may give different results; thus it is inevitable that a description of the properties of a randomized algorithm will involve probabilistic statements. For example, even when the input is fixed, the execution time of a randomized algorithm is a random variable"* - Karp (1991).

Given this class of algorithms, the main goal of this paper is to bring its advantages to deep learning. But it is neither obvious not intuitive what the advantages of randomization actually are. Nevertheless, in the last decades of intensive studies their benefits are established. To cite Karp (1991) (1989) again:

*"By now it is recognized that, in a wide range of applications, randomization is an extremely important tool for the construction of algorithms. There are two principal types of advantages that randomized algorithms often have. First, often the execution time or space requirement of a randomized algorithm is smaller than that of the best deterministic algorithm that we know of for the same problem. But even more strikingly, if we look at the various randomized algorithms that have been invented, we find that invariably they are extremely simple to understand and to implement; often, the introduction of randomization suffices to convert a simple and naive deterministic algorithm with bad worst-case behavior into a randomized algorithm that performs well with high probability on every possible input."*

Although much more manifold, one of the major advantages of randomized algorithms, which we also leverage to define a learning objective in the main text, is their robustness against adversarial inputs. To cite Karp (1991) a final time:

*"A game-theoretic view is often useful in understanding the advantages of a randomized algorithm. One can think of the computational complexity of a problem as the value of certain zero-sum two-person game in which one of the players is choosing the algorithm and the other player, often called the adversary, is choosing the input data to foil the algorithm. The adversary's payoff is the running time of the algorithm on the input data chosen by the adversary. A randomized algorithm can be viewed as a probability distribution over deterministic algorithms, and thus as a mixed strategy for the player choosing the algorithm. Playing a mixed strategy creates uncertainty as to what the algorithm will actually do on a given input, and thus makes it difficult for the adversary to choose an input that will create difficulties for the algorithm."*

Generally, one distinguishes between two classes of randomized algorithms, see below, whereas we focus on Monte-Carlo algorithms in this paper i.e. randomized algorithms with fixed runtime (such as a common feed-forward neural network) which we allow to produce incorrect predictions. We go on to furthermore discuss when and how one can boost the performance of these algorithms by repetition, a technique that we leverage heavily in the main text.

To showcase the advantages of randomized algorithms and give the reader a quick overview, we furthermore introduce and describe in the following a few classic randomized algorithms, which also served as an inspiration for deciding on which problems we trained neural networks on.

### E.1   TWO CLASSES OF RANDOMIZED ALGORITHMS: MONTECARLO VS. LASVEGAS ALGORITHMS

**Definition 3.** *A Monte Carlo algorithm is a randomized algorithm that runs for a deterministic runtime and whose output may be incorrect with some (usually small) probability.*

**Definition 4.** *A Las Vegas algorithm is a randomized algorithm that runs for a randomized runtime and always outputs a correct solution.*

These two notions are nevertheless tightly intertwined. Indeed, given a Las Vegas algorithm, one can construct a Monte Carlo algorithm by running the algorithm for a prefixed amount of time (i.e.

early terminating the algorithm if it exceeds this runtime) and returning whatever output if it didn't succeed. A simple Markov inequality allows to bound the probability of returning a wrong answer. In general however, a Monte Carlo algorithm cannot be converted into a Las Vegas algorithm. There are however special cases where this can be done. When for example the correctness of a solution can be tested with a deterministic algorithm, one can construct a Las Vegas algorithm from a Monte Carlo algorithm, by running it as many times as needed to find a correct answer. We describe this procedure in detail in the following section which we leverage heavily in the main text, denoted as *majority voting*, to boost performance.

### E.1.1 PROBABILITY AMPLIFICATION

Assume we have access to a randomized algorithm that can, for every single input $x \in \mathcal{X}$, output $A(x)$ that is correct with a certain probability. Note that in our experiments of the main text, the output of the neural network $A(x)$ returns a probability distribution e.g. the probability over certain classes or actions. We stress that to compute the final result of the algorithm we must therefore either sample to obtain the final result of the algorithm or compute it by the argmax. Note that this additional sampling / argmax step is usually not necessary for common (randomized) algorithms. In the following, we will show a procedure that allows to define another algorithm that will significantly amplify the probability of success of $A$. Therefore, we wish to explain the presented results in the main text where we observe majority voting of the randomized transformer predictions indeed improves performance significantly while surpassing the deterministic transformer models dramatically.

More precisely, let $\mathcal{Y}$ be the output space, and $C_x \subset \mathcal{Y}$ be the correct outputs for input $x \in \mathcal{X}$.

Let $x \in \mathcal{X}$. $A(x)$ is a distribution over $\mathcal{Y}$ (one can think of $A$ as the random output for a given input). We sample $N \in \mathbb{N}^*$ samples of $A(x)$ and return the statistical mode $M$, i.e., the most frequent output. We want to bound the probability to return a correct output $y \in C_x$.

**Proposition 2.** *Let $\delta > 0$,*

$$\Delta := \max_{y \in C_x} \mathbb{P}[A(x) = y] - \max_{y \notin C_x} \mathbb{P}[A(x) = y]$$

*If $\Delta > 0$ and $N > \frac{2}{\Delta^2} \ln \frac{\delta}{|\mathcal{Y} \setminus C_x|}$, then $\mathbb{P}[M \in C_x] > 1 - \delta$.*

The proof follows from Hoeffding inequality and a union bound.

We now present well-known randomized algorithms which should give the reader better intuition how and why randomness is used to design powerful algorithms.

### E.2 RABIN-MILLER (MILLER (1975); RABIN (1980))

**Setting**: Given a number $n \in \mathbb{N}$, decide whether $n$ is prime or not.

**Approach**: If $n$ is prime, any $a \in [1, \ldots, n-1]$ satisfies $a^{n-1} \equiv 1 \mod p$. We can write $n - 1 = 2^v m$ where $m$ is odd. We necessarily have that $a^{\frac{n-1}{2}} \equiv -1 \mod p$ or $a^{\frac{n-1}{2}} \equiv 1 \mod p$. By recurrence, we either have $a^m = 1$ or there exists a $k < v$ such that $a^{2^k m} \equiv -1 \mod p$. One can show that, if $n$ is composite, at most $\frac{1}{4}$ of the $a$ in $[1, \ldots, n-1]$ will verify this property.

A naive solution would be to then try all possible bases. That would give an exact but very inefficient algorithm. Instead, we can sample independently and uniformly at random $k$ bases, and return "pseudoprime" if all tests are successful, and "composite" otherwise. If the algorithm gets a prime, it will always output "pseudoprime". If it gets a composite number, it will output "pseudoprime" with probability $\frac{1}{4^k}$. Conversely, the number is always composite if the algorithm returns "composite". Using Bayes rule, one can easily bound the probability that the number is composite given that the algorithm returns "pseudoprime". The complexity of the algorithm is $\tilde{\mathcal{O}}(k \log^2 n)$, much less than the $\tilde{\mathcal{O}}(n)$ we would have needed with the deterministic approach.

Assuming the Grand Riemann hypothesis, it is enough to test the first $\mathcal{O}(\log^2 n)$ basis to get an exact deterministic algorithm running in $\tilde{\mathcal{O}}(\log^4 n)$. The hypothesis remains an open problem so far. The randomized version is still mainly used in practice because the deterministic algorithms that exist are much more complex mathematically, more difficult to implement, and more expensive in terms of computational complexity.

### E.3 PAGING PROBLEM (SLEATOR & TARJAN (1985B))

**Problem**: We are given a **fast memory** (cache) that can fit $k$ pages and an unlimited **slow memory**. A user can only access data from the cache while there is communication channel between the fast and the slow memory. If a user wants to access data which is not currently in cache, it requires a cost of 1 to overwrite a cache entry by the required page from the slow memory. Writing data from the cache to the slow memory costs 0. Therefore the goal of this problem is to find a strategy to fetch as little data from the slow memory. An oblivious adversary chooses a sequence of queries. What is the best online memory management strategy to minimize the cost?

**Approach**: One of the easiest strategies, the Random Marking Algorithm, goes as follows: initialize all pages in the cache as marked; whenever the queried page is not in the cache, evict one of the unmarked cache pages chosen uniformly at random; in case all pages are marked, unmark all before. It can be shown that this achieves is $\mathcal{O}(\log k)$-competitive against an oblivious adversary. This is also the best one can achieve theoretically. One can show that any deterministic algorithm can at best be $k$-competitive against an oblivious adversary.

### E.4 CYCLE 3-COLORING

**Problem**: Given an oriented cycle defined by a set of $N$ vertices $V = \{v_i\}_{i=1}^N$ and edges $E = \{(v_i, v_{i+1 \mod N})\}_{i=1}^N$, color the vertices using 3 colors such that neighbouring vertices receive different colors. Formally speaking, we desire a mapping $f : V \mapsto \{1, 2, 3\}$ such that $f(v) \neq f(w)$ for all $(v, w) \in E$. In the distributed graph coloring problem, which we consider here, we require that all vertices $v$ compute $f(v)$ locally, being able to only communicate with their immediate vicinity.

**Approach**: A simple randomized approach proceeds in two phases: In the first phase, each vertex selects itself with probability 1/2. If one of its neighbours also selected itself, it subsequently unselects itself again following some deterministic rule based on the vertex id. A typical example for such a rule, is that always the one with smaller id unselects itself. In the second phase, every vertex that is still selected colors itself with color 0. Crucially, one easily shows that, regardless of the distribution of the vertex ids, this process ensures that with high probability the paths between vertices in color 0 have lengths at most $\mathcal{O}(\log n)$. Therefore, they can be colored properly, in a third and final phase, by a simple propagation process using only $\mathcal{O}(\log n)$ rounds. This defines a Las Vegas algorithm, that one can turn to a Monte Carlo algorithm (check section E.1). Using a different approach, one can actually achieve a much better complexity of $\mathcal{O}(\log^*(n))^\dagger$ (Rybicki & Suomela (2015)).

---

$^\dagger \log^*(n)$ counts the number of times one has to apply log composedly to become smaller than 1

