# OpenReview forum: "Learning Randomized Algorithms with Transformers"
_ICLR.cc/2025/Conference — ICLR 2025 Oral_

### Official Review · Reviewer_BfRQ · 2024-11-01

**Soundness:** 3
**Presentation:** 4
**Contribution:** 3
**Rating:** 6
**Confidence:** 3

**Summary:**

This paper explores the integration of randomization into transformer models, traditionally used in deterministic settings, to enhance their performance, especially in adversarial contexts. The authors demonstrate that randomized algorithms can be instilled in transformers through learning, in a purely data- and objective-driven manner. Though an analysis of known adversarial objectives for which randomized algorithms offer a distinct advantage over deterministic ones, they show that common optimization techniques, such as gradient descent or evolutionary strategies can effectively learn transformer parameters that make use of the randomness provided to the model. To illustrate the broad applicability of randomization in empowering neural networks, we study three conceptual tasks: associative recall, graph coloring, and agents that explore grid worlds.

**Strengths:**

- The paper is very well written, clear, with a good motivation. They first introduce the setting by presenting an example with associative recall, which is a variant of the classical paging problem in computer science for which randomized solutions exist. This problem is also a well-studied problem for transformers with the goal of evaluating recall capabilities. The paper continues with a theoretical analysis in which the authors study when randomization can be beneficial and when it is not. From there, the authors propose a training objective.
Finally, the authors experiment with 3 use cases: associative recall, graph coloring, and agents exploring lattice worlds.
- The contribution is, to the best of my knowledge, novel and could lead to improvements in how transformers can solve certain tasks.

**Weaknesses:**

- The experiments are performed on three different small tasks, the proposed approach should increase the computational complexity, which is already high with transformers, can the authors comment on this in terms of training and inference?
- I am a bit puzzled by the paragraph on adversarial robustness (small input perturbations) in the related work, first, the cited paper (Rakin et al. (2018)) shows an approach to increase robustness with randomization, but now such approaches have been mostly disproved (see Gnecco et al. 2023). Furthermore, the randomized smoothing approach uses randomization _only in the training phase_, the approach is inherently a training procedure for a smooth function. The inference _is_ deterministic. Overall, I find this discussion could be made clearer.

Gnecco et al. On the Role of Randomization in Adversarially Robust Classification. NeurIPS 2023.

**Questions:**

See first Weakness

---

> ### Author Response · Authors · 2024-11-22
>
> We thank the reviewer for the constructive comments that have helped us improve the clarity of our paper. We reply to each point separately below.
>
> Weakness 1) Please see our general response above. We remain open to discuss any further issue the reviewer might have.
>
> Weakness 2) Thank you for pointing out this related work which we were not aware of. We clarify that randomization as an approach to improve robustness has been disproved in restricted settings (such as the 0-1 loss in the binary classification setting with specific hypothesis classes considered in Gnecco et al. 2023), but it has not been disproved in most practical settings. Furthermore, we emphasize that we are not advocating for using randomization at inference time - in fact, the majority vote inference which uses properly randomized models is a prime example of a deterministic inference. Rather, we believe that given a model architecture, achieving proper randomization is useful for various reasons, including for boosting robustness by flexibly allocating additional computation at inference time as we show in our paper. This is in fact aligned with the theoretical result of Gnecco et al. 2023, as well as existing approaches such as randomized smoothing.
>
> We will clarify the paragraph by incorporating the known limitations of randomization as a strategy to improve robustness (Gnecco et al. (2023)) to better position our setting. We will also improve the presentation of the paragraph by better distinguishing existing procedures that aimed at properly randomizing models during training  (e.g. Rakin et al. (2018)), and those that aimed at leveraging randomized models to improve inference.
>
> We thank the reviewer again and remain open to discuss any follow up question. We hope our clarifications could address your concerns, and can convince the reviewer to vote for a strong acceptance of the paper.

---

> > ### Comment · Reviewer_BfRQ · 2024-11-26
> >
> > Thank you for the response. I keep my positive score.

---

### Official Review · Reviewer_mQSR · 2024-11-02

**Soundness:** 3
**Presentation:** 4
**Contribution:** 3
**Rating:** 6
**Confidence:** 3

**Summary:**

The paper explores how transformers can learn and implement randomized algorithms, which are advantageous in certain adversarial and game-theoretical contexts. Randomized algorithms typically perform well in adversarial environments and exhibit robustness against worst-case scenarios. The authors propose a novel objective function for training transformers that optimizes their performance in these settings by introducing randomization via an input "seed."

**Strengths:**

- The authors provide a strong theoretical foundation for why randomization is advantageous in certain adversarial contexts, referencing game theory and established concepts like Yao’s Minimax Principle.
- The study highlights that randomization can increase resilience against adversarial attacks.

**Weaknesses:**

- While the paper presents conceptual tasks to validate the approach, it does not provide empirical results on large-scale or real-world datasets. This limitation raises questions about how well the method would scale and perform in more complex, realistic environments.
- The approach requires sampling multiple seeds during training (controlled by the hyperparameter mm), which can increase computational overhead significantly. The authors note that this limitation affects memory, training time, and inference time, making the method potentially impractical for large-scale applications.
- The paper’s theoretical sections are rigorous but may be dense for readers without a background in adversarial training or randomized algorithms.
- The study assumes relatively static environments for tasks like grid world exploration and graph coloring. However, it does not address how the transformer’s randomized strategies would perform in dynamic, non-stationary settings where environmental conditions change over time.

**Questions:**

- What are the computational and memory challenges of scaling the proposed randomization method to larger, more complex datasets or real-world tasks? Are there specific optimizations that could make the approach more efficient?
- The study introduces hyperparameters qq and mm in the training objective. How sensitive are the final model’s performance and robustness to variations in these hyperparameters?
- Could the proposed randomization approach be adapted for more complex reinforcement learning environments, such as those with continuous action spaces or requiring long-term strategy?
- How does this approach compare with other randomization techniques, such as dropout (at inference)?

---

> ### Comment · Reviewer_mQSR · 2024-11-18
>
> Dear authors,
>
> while reading the other comments I wanted to specifiy a bit more concret some of my comments:
>
> - "While the paper presents conceptual tasks to validate the approach, it does not provide empirical results on large-scale or real-world datasets. This limitation raises questions about how well the method would scale and perform in more complex, realistic environments."
> --> About the datasets I was thinking about to use the algorithms and try to do some banchmark tests on some algorithm competions sides.
>
> - "The paper's theoretical sections are rigorous but may be dense for readers without a background in adversarial training or randomized algorithms."
> --> Section 2.3 RANDOMIZATION CAN BE BENEFICIAL IN ADVERSARIAL SETTINGS looked a bit short to me.
>
> - "The study assumes relatively static environments for tasks like grid world exploration and graph coloring. However, it does not address how the transformer's randomized strategies would perform in dynamic, non-stationary settings where environmental conditions change over time."
> --> How would it perform dynamic graph coloring? Are dynamics a limitation?
>
> - "The approach requires sampling multiple seeds during training (controlled by the hyperparameter m), which can increase computational overhead significantly. The authors note that this limitation affects memory, training time, and inference time, making the method potentially impractical for large-scale applications."
> --> Some numbers (runtime, relative performance per operation) etc. would be good to have. Also if one has a "optimal" algorithms known how much more operations are need with the transformer approach?

---

> > ### Author Response · Authors · 2024-11-22
> >
> > We very much thank the reviewer for this in-depth review and the time they spend evaluating our work. We answer the questions one-by-one.
> >
> > Weakness 1 & 2 and additional comments 1 & 4) Please see our general comment on the scalability of our approach in the general response, we remain open for a discussion if there are any further questions.
> >
> > Weakness 3) We thank the reviewer for bringing up this point. We actually put in considerable effort to i) help building up intuition and ii) explaining the theoretical insights and results. Could the reviewer give more concrete suggestions on what exactly it is that is unclear and can be improved? We want to make these results as accessible as possible, so these comments are very valuable to us. Thank you!
> >
> > Weakness 4) Thank you for raising this interesting point. Although we think that it is out of scope for the current work, we very much agree that moving to more real-world non-stationary problems is an important next step, see our general response.
> >
> > Q1) Please see our general comment on the scalability of our approach in the general response. We are open to discuss any follow up questions the reviewer might have.
> >
> > Q2) Please see Figure 2 left, as well as Figure 3 two outer right plots, as well as Figure 5 two outer left plots as well as Figure 7 left plot for analyses studying what happens if you change $q$ and $m$ in all 3 of our tasks.
> >
> > Q3) We don’t see any constraints of not applying our approach in RL with continuous action spaces or problems that require long-term strategies. Stochasticity in actions can be naturally obtained in common reinforcement learning settings or autoregressive models with discrete outputs, through sampling from the softmax. This is not as straightforward for models with continuous outputs. Interestingly, we believe that our method actually provides an attractive way to obtain stochasticity in models of continuous outputs, without any modifications to the models, but leave this direction for future point. Thank you for raising this point which we will discuss in the revised document.
> >
> > Q4) We believe that our method i.e. the training objective we provide is orthogonal to the way noise is inserted in the model. We could actually use dropout or additive noise to the weights or activations as the source of noise. We chose input noise so the noise could be easily ignored by the model, as shown by experiment when using $q=1$. This, we argue, is significantly more difficult for neural network models where noise is provided inside the architecture.
> >
> > Additional comment question on dynamic graph coloring - We believe that our objective is general and can lead to powerful randomized algorithms, when the discussed assumptions are met. Therefore, we do not believe that our approach is, in principle, not able to handle more online (graph coloring) settings.
> >
> > We thank the reviewer again and remain open to discuss any follow up question.

---

> > > ### Comment · Reviewer_mQSR · 2024-11-25
> > >
> > > Dear Authors,
> > >
> > > **Weakness 1 & 2 and additional comments 1 & 4)**: I am fine with your answers thank you.
> > >
> > > **Weakness 3)**: See my comments.
> > >
> > > **Weakness 4)**: Fine with this.
> > >
> > > With the rest I am also fine! Thank you for the answers. I will keep my score.

---

### Official Review · Reviewer_pYUp · 2024-11-02

**Soundness:** 3
**Presentation:** 2
**Contribution:** 2
**Rating:** 6
**Confidence:** 4

**Summary:**

The paper explores the incorporation of randomized algorithms into transformer models through learning, in order to enhance robustness and performance in adversarial environments. By training transformers on expected and adversarial losso, single and multiple seeds, the study demonstrates that their proposed approach can outperform their deterministic counterparts, particularly when faced with adversarial challenges. The authors focus on three main tasks: associative recall, graph coloring and exploration in grid worlds, to evaluate the impact of randomized strategies on model performance. The results indicate that, by employing randomness, transformers achieve higher robustness and adaptability (implementing simple strategies repetition or majority voting) over predictions with different random seeds.

**Strengths:**

1) Innovative integration of randomization: Altough randomization was a concept that has been used with Transformers during the years and at different levels (e.g. positional encodings, attention weights) the paper introduces an original concept of definining randomized algorithms within neural networks through learning: by simple epmloying repetitions and strategies such as majority voting this approach outperforms deterministic approach

2) Comprehensive experimental design: The paper provides a solid experimental framework across varied tasks, such as associative recall, graph coloring, and grid world exploration. Each task is carefully selected to illustrate different advantages of randomization, such as memory management, combinatorial problem-solving, and exploration.

3) Theoretical justifications and adversarially driven objective : The authors define a description of theoretical considerations about randomized algorithms that bring to the definition of an adversarially driven objective: it is an interesting addition that supports the study's claims on robustness. The use of a relaxed adversarial loss and exploration of different adversarial strengths adds depth and completeness to the analysis of transformer behavior in challenging environments.

4) Empirical results on robustness: The empirical results are thorough and show randomized transformers advantages, eespecially in scenarios with worst-case inputs. The paper's analysis highlights how majority voting and the number of seeds influence performance, enhancing the robustness of the findings. Moreover tha authors discuss about their results and the major limitations of their approach.

5) Appendix: Further information on the training modality and parameters is appreciated.

**Weaknesses:**

1) Scalability challenges: The most important corcern about the proposed methodology is the computationally cost, particularly with the reliance on multiple seeds and adversarial loss training. The authors acknowledge this in the "Summary and Limitations" paragraph, noting that scaling the approach to larger settings may require significant computational resources, which limits the practicality and broader applicability of the approach. I consider that such a problem should have been addressed in a more in-depth manner and not relegated to a second analysis, as it is a very important measure for judging the entire methodology.

2) Limited practical applications discussed: While the theoretical foundation is strong, the paper could benefit from discussing more real-world applications where randomized transformers might be beneficial.

3) Dependency on the choice of q and m hyperparameters: The model's performance and randomization effectiveness heavily depend on hyperparameters, particularly q and m. The authors could delve further into automated ways to tune these values or provide guidelines for choosing optimal parameters for various tasks. High-computation hardware needed to perfom analysis about the hyperparameters can be prohibitive.

**Questions:**

What were the reasons that pushed the authors not to present analyses (even theoretical) regarding the computational cost preferring to insert more general information such as section E of the appendix?

---

> ### Author Response · Authors · 2024-11-22
>
> We thank the reviewer for this positive and detailed review! We answer the questions one-by-one.
>
> 1) Please see our general response addressing scalability challenges.
>
> 2) Thank you for bringing up this important point. Please see again our general response addressing scalability challenges but we highlight here that we will add a discussion of real-world and future applications in the revised manuscript.
>
> 3) Thank you for raising this point. Any general hyperparameter search techniques should be applicable to $m$ and $q$ as well. On the other hand we speculate that generally, the choice of $m$ is actually not particularly sensitive and can be adjusted to computational constraints, as long as it is above 1. Too small $q$ on the other hand can, as our results show, lead to suboptimal randomized algorithm performance. Nevertheless, we do not see any downside of choosing moderate, for example $q=10$, when assuming multiple forward passes during inference time such that majority voting improvements can be obtained. Concluding: Although the hyperparameters are important and task dependent, we believe that default values of $q=10$ and $m>1$ will usually, as we observed, lead to powerful randomization.
>
>
> We thank the reviewer again and remain open to discuss any follow up question.

---

> ### Comment · Reviewer_pYUp · 2024-11-27
> **Thanks**
>
> I appreciate the authors response, which has addressed all of my questions. I will maintain my previous score.

---

### Official Review · Reviewer_y3S8 · 2024-11-04

**Soundness:** 3
**Presentation:** 3
**Contribution:** 3
**Rating:** 8
**Confidence:** 4

**Summary:**

Randomized algorithms can offer better worst-case performance than deterministic ones.
Can transformers learn randomized algorithms?
The authors study conditions for randomized algorithms to yield possible advantages and they offer a training methodology to emerge them in transformer networks. Namely design choices to emerge randomized algorithms involve around: a. the model; it needs to have limited capacity and receive random signal at its input, b. the loss function; it needs to be a surrogate for worst-input risk instead of expected risk.
They demonstrate application in three conceptual tasks: associative recall, graph coloring, agent exploring a grid-world
And they claim that randomized algorithms achieve increased robustness against adversaries.

**Strengths:**

Very well-written paper, easy to follow, with extensive examples and related work material both in the main paper and in the appendix.
Well-motivated and executed study for the possibility to learn randomized algorithms.

1. Convincing results of superior worst-case performance in the considered tasks when compared to training alternatives, ablating some aspect of the loss, the multi-seed version trained on the relaxed adversarial loss is the best.
2. Experiments cover well important hyperparameters for the training setting. namely the relaxation hyperparameter which interpolates between ERM and worst-case risk, $q$, and the number of seeds per input, $m$.
3. Experiments cover variety of architectural constraints, such as linear attention (sec 3.2), structured local attention masks that are GNN-like (sec 3.3), and autoregressive settings (sec 3.4). As well as a variety of tasks and optimization techniques.

The paper finds the reader with a clear impression about the possibility of randomized algorithms for improving worst-case performance, as evidence by 95th percentile results in Figs 2,3,5 and 7. As well as inference-time strategies on improving them with majority voting.

**Weaknesses:**

1.a. I would personally have enjoyed seeing experiments with actual adversarial robustness training, using a threat model on human preference alignment data, applied at generative LLMs.

1.b. Lines 216-218 mention that adversarial robustness training is difficult, however recent literature has provided with framework for that [ 1]. While it does not decrease the contributions of this paper, comparing with the methodology mentioned in the paper would have been great!

[1] Xhonneux et al., “Efficient Adversarial Training in LLMs with Continuous Attacks”, May 2024 to appear at NeurIPS 2024.

2. Section 2.1 “Excessive Model Capacity Will not Enforce Randomness” the argument is not very convincing to me. In the face of a multi-seed worst-case objective like one described later, does the argument still hold? Proposition 1 also seems to go against the impossibility eluded by this section. Experiments on finetuning medium-sized (~1B) models on toy tasks could have provided some extra evidence for that.

3. On grid-world experiments: While the first episode demonstrates randomness in ways to explore for the target, why doesn’t the randomness follow as well in the second “exploitation” episode? There are multiple equivalent in length ways to reach the target. I would expect that in Figure 6.C. those would vary across different seeds.

4. Instead of using input randomness, it would have been very interesting to understand the ability or limitations on using output sample randomness which is fed back to the input in an autoreregressive case, as a drive to learn randomized algorithms. The authors avoid this explicitly in Section 3.4, however I do not understand the reason.


## Typos
L141: “for any distribution **over** $\mathcal{X}$ and $\mathcal{R}$.
L233: “hyperparameters $m$ and $\mathbf{q}$.
L965, L895, L853: change $p$ for $q$.


## Missing citations
On linear attention (L291): Katharopoulos et al., “Transformers are RNNs: Fast Autoregressive Transformers with Linear Attention”, ICML 2020.

**Questions:**

1. The transition to use the q-norm relaxed loss in L195-200 is not motivated sufficiently at that point in text. Can you elaborate on it a bit more?

2. L207-208: Are the same random input seeds reused for all inputs? Why? What would happen if fresh seeds are sampled for every input $x_i$?

3. Batching strategy when finetuning for the emergence of a randomized algorithm is not clear at the text. I assume that $B \times N$ samples are processed independently at once. $B$ is the number of input data, $N$ is the number of random seeds. Is this correct?

---

> ### Author Response · Authors · 2024-11-22
>
> We thank the reviewer for the positive and encouraging review. We address their comments one by one.
>
> Weakness 1) a) Please see our general response addressing scalability, and we agree with the reviewer that experiments using adversarial robust training in combination with our randomization strategy, for example in conjunction with the mentioned efficient adversarial training strategy, is a very promising idea but out of scope for this work.
>
> Weakness 1) b) Note that the lines 216-218 refer to equation 2 i.e. a general min-max loss where in generality one can not compute gradient attacks wrt to e.g. the inputs to efficiently find an attack. In cases such as [1], this is simplified by using continuous attacks, which can be turned into discrete attacks but still fall, we believe, out of the scope of the current work. We will clarify the text accordingly, thank you for pointing out this source of confusion.
>
> Weakness 2)  We thank the reviewer for raising this important point. We stress that the assumptions in Proposition 1 do contain the necessary condition for randomization to present a benefit raised in paragraph 2.1. Indeed, if the model were overparameterized w.r.t the training set, the model would be able to perfectly fit the data, i.e. achieve 0 loss on all training data points. This in turns would imply that for any seed $r_i$, the worst case data is the set of all training data (i.e. $\arg\max_{x'\in\mathcal{X}} L(A_{\theta^*}(x',r_i)) = \mathcal{X}$), which would violate the second condition in equation 3. On a more intuitive level, if the model has enough capacity to perfectly fit the training set without leveraging the random seed encodings, then there is no incentive in properly leveraging randomness. In principle, a lack of incentive in terms of optimization does not mean that randomness won't in fact be leveraged. However, in that case no theoretical guarantee can be made about proper randomization at a global optimum of the objective.
> We hope we clarified this important point, thank you again for raising it, and remain open if you have any further questions.
>
> Weakness 3) This is a very good point indeed. The reviewer is right that there are several paths of equal optimality in the second episode. However, there is no incentive for the model to not choose a single path, as the incurred losses are all equal. This highlights the crucial point that, simply because randomness is available to the model, the model will not necessarily make use of it. In fact, we hypothesize that due to e.g. the necessity for richer internal representation, it is in practice necessary that randomness be incentivized by a lower loss for proper randomization. We will comment on this in the revised manuscript, a very good observation that is important to discuss.
>
> Weakness 4) The source of randomness is indeed an interesting question but we decided to have a simple source of randomness across all of our settings. Nevertheless, we indeed do not see any reason why in principle noise stemming from sampling actions in the autoregressive case could not be leveraged to learn randomized algorithms. We will clarify this in the text.
>
> Question 1) We can indeed make this clearer. Intuitively the q-norm allows to account for a continuum of differentiable losses with different behaviors. For q = 1 we have the ERM loss $L^E$ while for $q \to \infty$ we have the adversarial loss $L^A$. While one would want to optimize for $L^A$, this would end up accounting for only one $x$ for an expense of computation the loss for a batch of input points, resulting in a slow and high variance training. In practice, it is better to relax this loss by scaling” the losses of different points as a function of their magnitude. This is what the $L^q$ allows to achieve.
>
> Question 2)  This is an excellent point. In our experiments, we have indeed been using the same random seeds for all inputs. Let’s first observe that whether we use fresh seeds or the same seeds, the resulting estimator of the loss $L^q $, which we aim to optimize, remains biased. Importantly, the bias of the gradients in both cases are comparable, as they align perfectly in expectation as the expected gradients are the same up to renormalization. Second, according to the strong law of large numbers, the estimators will almost surely converge to $L^q $ in either case. This aligns with both our intuition and our experimental results.
>
> However, while the biases remain comparable, the variance can differ. It is mathematically possible to construct scenarios where one choice provides a better estimate than the other. Initially, we experimented with decorrelated seeds but found that using correlated seeds improved the outcomes, prompting us to adopt this approach.
>
> Question 3) This is correct.
>
> We thank the reviewer again for this detailed and very helpful review! Thank you for spotting the mentioned typos.

---

> > ### Comment · Reviewer_y3S8 · 2024-11-25
> > **Thank you**
> >
> > I thank the authors for their detailed response, which clarifies all of my questions, I am looking forward to see these clarification in text as well. I am keeping my original positive score.

---

### Author Response · Authors · 2024-11-22
**General Response about real-world applications and scalability**

We want to thank all reviewers for their time and effort in evaluating this work and the generally positive reviews. We address here in a general answer the concerns about the scalability of the approach.

1) We want to highlight that the goal of this paper is to build a theoretical understanding, backed by empirical evidence, providing insights under which assumptions neural networks, through learning, become randomized algorithms. We therefore focused on studying well known problems from the randomized algorithms literature, for which powerful randomized algorithms are known and well studied.

2) Although the proposed objective does introduce an overhead compared to common empirical risk minimization (ERM), other related adversarial training objectives can also be considerably more expensive than ERM. For example, common gradient based adversarial training techniques require up to 10-100x more compute, see for example Table 1 of Madry et al., 2018. Furthermore, we stress that in all of our experiments rather small $m=10$ (leading to 10x more compute) led to near optimal performance, when compared to larger $m>10$.

3) We want to put forward more comments and ideas on how to address scalability issues with the current approach:
  - Fine-tuning: We find an interesting future research avenue to investigate if pre-trained models can be randomized by fine-tuning on our objective.
  - A relaxation of this abrupt 1) pre-training and 2) fine-tuning stages, would be to gradually increase $m$ and $q$ of our objective during training. In the beginning of training, the model could be trained as usually done via ERM (q=1, m=1) and then gradually increased to enforce randomization e.g. q=10, m=10. We argue that this is a scalable and intriguing future direction.

4) We agree with the reviewers, and will highlight this in the revised manuscript, that other objectives, leveraging gradient based attacks, could exchange the arguably expensive  $m$ forward passes, which can easily be parallelized, with the costs of finding attacks by gradients. We agree that this is a promising future direction in order to scale our approach to real world settings. We will add, at least in the appendix if space does not permit, a discussion on real-world applications and possible future directions.
Here, we chose our training objective, the relaxation of the min-max objective through the q-norm, to allow differentiability with respect to the parameters while tackling problems where differentiating with respect to the inputs, and therefore allowing for gradient based attacks, is not possible. Note that this situation is common for problems in field of randomized algorithms, reflected by the tasks that we study in our work.

We again want to express our sincere gratitude for this set of high quality and insightful comments and suggestions which benefited the quality of the work greatly. We welcome any further suggestions and questions.

---

### Comment · Area_Chair_bGUJ · 2024-11-25

Dear reviewers,

A reminder that **November, 26** is the last day to interact with the authors, before the private discussion with the area chairs. At the very least, please acknowledge having read the rebuttal (if present). If the rebuttal was satisfying, please improve your score accordingly. Finally, if you have concerns that might be solved in time, this is the last chance before moving on to the next phase.

Thanks,
The AC

---

### Meta-Review · Area_Chair_bGUJ · 2024-12-15

**Metareview:**

The paper considers transformers endowed with a further random seed as input, which are used to learn randomized algorithms. They show (both theoretically and empirically) that this is beneficial in adversarial situations, where it is known from standard computer science that the added randomness "breaks" the adversary attack.

The paper had 4 reviews, all of which were positive on the contributions and recommended acceptance. The authors provided a very strong rebuttal which clarified several points, and the reviewers maintained their positive score.

In general, all reviewers praised the paper's writing, its contributions, and the novelty. Their concerns were generally aligned (e.g., computational complexity, as described below) but well addressed in the rebuttal. No concerns remained after the rebuttal.

In summary, the paper is a very strong contribution describing an innovative framework for exploiting randomness in learning algorithms with transformers, which may have an impact also outside the scope of algorithmic reasoning. I see no reason to override the reviewer's consensus and I strongly recommend acceptance.

**Additional Comments On Reviewer Discussion:**

- **Reviewer BfRQ** was concerned about computational complexity (as the method requires calling the forward pass of the transformer multiple times) and suggested a few additional related works. In general, they praised the novelty, contribution, and writing of the paper.

- **Reviewer mQSR** was also concerned about computational complexity, as well as the applicability of the method to real-world scenarios and dynamic environments. The rebuttal answered most concerns.

- **Reviewer pYUp**, similarly, was concerned about computational complexity, lack of real-world scenarios, and dependence on two hyper-parameters. Also in this case, the rebuttal was positive.

- **Reviewer y3S8** asked a few clarifications and also suggested additional experiments with human preference data and adversarial robustness training.

All reviewers are aligned on the paper and they weighted similarly in my evaluation.

---

### Decision · Program_Chairs · 2025-01-22

Accept (Oral)